# A Chinese Multimodal Social Video Dataset for Controversy Detection

## ABSTRACT

Social video platforms have emerged as significant channels for information dissemination, facilitating lively public discussions that often give rise to controversies. However, existing approaches to controversy detection primarily focus on textual features, which raises three key concerns: it underutilizes the potential of visual information available on social media platforms; it is ineffective when faced with incomplete or absent textual information; and the existing datasets fail to adequately address the need for comprehensive multimodal resources on social media platforms. To address these challenges, we construct a large-scale Multimodal Controversial Dataset (MMCD) in Chinese. Additionally, we propose a novel framework named Multi-view Controversy Detection (MVCD) to effectively model controversies from multiple perspectives. Through extensive experiments using state-of-the-art models on the MMCD, we demonstrate MVCD's effectiveness and potential impact.

## CCS CONCEPTS

• **Information systems** → **Information systems applications**;
• **Social and professional topics** → **Professional topics**; • **Computing methodologies** → **Artificial intelligence**; **Machine learning**.

## KEYWORDS

Controversy Detection, Dataset Construction, Social Video Platform

## 1 INTRODUCTION

With the prevalence of social video platforms, videos have become an important information-sharing channel. Videos uploaded on social media platforms quickly accumulate thousands of views within seconds, facilitating worldwide user engagement in opinion shaping [40]. However, the openness of these social platforms also gives rise to fervent discussions and the exchange of divergent opinions. Consequently, the proliferation of numerous videos necessitates implementing risk management and control measures. Our work focuses on controversy detection, which serves as the basis for exploring various advanced applications such as risk indication and brand reputation management. Previous research on controversy detection has primarily focused on the textual modality, neglecting the necessity of incorporating multimodal in situations where textual information is limited. In light of the ubiquity of social video

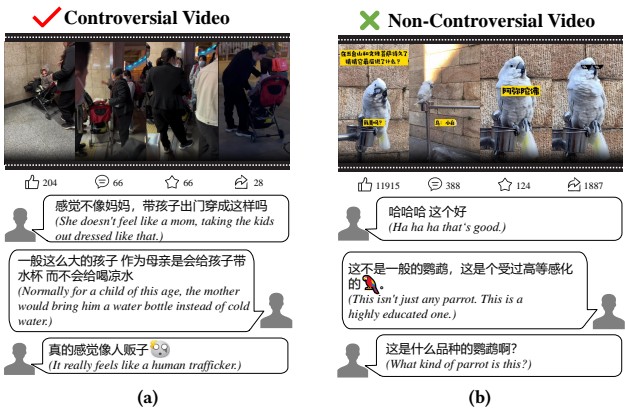

**Figure 1: Examples of controversial and non-controversial videos.**

platforms, we specifically concentrate on multimodal controversy detection. This involves the utilization of various modalities, including video, text, and metadata, to discern the controversy inherent in content.

Controversy exists as a form of public discourse, attracting an increasing number of opposing viewpoints and resulting in escalating divergence or polarization [10, 16, 36, 45]. As controversies stem from the beliefs and values of participants, the exchange of opinions goes beyond mere "facts" and evokes intense emotions [28, 37]. Certain individuals may find their rights infringed upon due to controversial videos, primarily manifested through verbal accusations from others or conflicts and dissatisfaction arising from them. While being under the spotlight, social media environment can potentially cause psychological harm to these individuals, and in extreme cases, even lead to radical actions. In particular, when addressing controversial videos that discuss policies related to specific interests or incite intense discussions on social media platforms, there exists an inherent potential for the emergence of broader public controversies in the future [12, 37]. Hence, controversy detection on social video platforms plays a crucial role in providing an indicator for assessing the contentious nature of videos. It is necessary to curb the spread of controversies and mitigate public opinion risks. Additionally, controversy detection can generate recommendations that promote a "healthier diet" on social media [21].

Inspired by previous research work [10, 16, 30, 36, 45], we define controversial videos on social media platforms by considering three aspects. The first aspect concerns whether the *video content itself is prone to controversy*, such as whether it contains sensationalism, violent information, and so on. The second aspect examines a *conflict between the video content and the users' comments*. Controversial videos are identified if the comments exhibit opposing viewpoints to the video, personal attacks against the video creator,

criticism of depicted phenomena, or criticism/questioning of individuals/objects featured in the video. The third aspect focuses on the *controversy within comments*, specifically looking for clear opposing viewpoints expressed through support and opposition. Figure 1 provides an example of a controversial video and a non-controversial one. Figure 1(a) depicts a video showing a woman pushing a baby in a stroller, displaying an apparent lack of experience in child care. The comments associated with the video express suspicion and criticism towards the individual portrayed, evoking intense emotions and indicating a certain level of controversy. On the other hand, Figure 1(b) showcases a comedic video featuring a Yiwu Mountain parrot, wherein the comments applaud the parrot's performance, conveying a more positive sentiment without any controversial elements.

Existing approaches for detecting controversy on social media have primarily focused on leveraging semantic and structural features of target posts and their comments. However, there are three critical concerns. First, they overlook the potential of utilizing visual features available on social media platforms. Second, the current models for controversy detection often underperform when confronted with incomplete or missing text. Lastly, existing datasets offer a limited number of instances and lack simultaneous information on video, text, and user profiles [2, 31, 54].

To bridge this gap in multimodal controversy detection datasets, we introduce Multimodal Controversy Detection Dataset (MMCD), a large-scale dataset in Chinese that encompasses video content and rich social context. The MMCD dataset offers abundant features, providing an opportunity to evaluate various approaches for controversy detection and facilitate a deeper understanding of controversy dissemination and potential interventions. To comprehensively analyze the characteristics of controversial videos, we conduct an exploratory analysis of MMCD from multiple perspectives, offering valuable insights into effective detection strategies.

Furthermore, to address the challenges associated with multimodal controversy detection, we propose Multi-view Controversy Detection (MVCD) framework and conduct extensive experiments to compare with existing methods on MMCD. Experimental results validate the superiority of our proposed framework. Additionally, ablation experiment results validate the effectiveness of each individual module and modality within the framework. Moreover, early predictions indicate that our proposed framework is capable of handling scenarios with limited availability of comments.

Our main contributions are summarized as follows:

- We have developed and released MMCD, a Multimodal Controversy Dataset in Chinese, providing a valuable resource for studying controversies. This dataset is derived from social video platforms and includes a wide range of video content accompanied by extensive social context.
- We have conducted a comprehensive analysis of the constructed dataset, providing insights and findings relevant to further research.
- We have devised a multimodal controversy detection framework, Multi-view Controversy Detection (MVCD), which effectively models multimodal video content and captures the interaction between social contexts, enhancing the accuracy of controversy detection.

- Extensive experiments using state-of-the-art methods have been conducted on the MMCD, showcasing the effectiveness of our proposed approach and shedding light on the inherent challenges associated with multimodal controversy detection. To facilitate further research, we have made our work publicly available, including the codebase[1].

## 2 RELATED WORK

### 2.1 Controversy Detection Datasets

Currently, controversial detection datasets focus primarily on the textual modality. Table 1 provides a comprehensive overview of available datasets, covering aspects such as feature, category, language, accessibility, source, and time span information. These datasets are largely derived from three primary sources: *web pages*, with a particular focus on Wikipedia [17, 31]; *news websites*, such as The Guardian[2], EMOL[3], and Toutiao[4], which have contributed valuable datasets in this domain [2, 30, 39]; and *social media platforms* including Twitter, Weibo, Reddit, and others, which have also served as substantial sources of controversy detection datasets [9, 11, 13, 48, 54]. However, we identify a noteworthy gap in the availability of datasets specifically tailored for social video platforms. These play a crucial role in information dissemination and are frequent generators of controversies [16, 21, 40]. Consequently, we propose the collection of multimodal data as an effort to bridge this gap and provide necessary resources for detecting controversies in social videos.

### 2.2 Controversy Detection Techniques

Early methods for controversy detection primarily relied on statistic-based approaches, which involved analyzing user edit history [47], revision time [26], and context information [18, 49, 53]. Furthermore, some researchers incorporated textural features, such as controversial vocabulary [14], sentiment [27, 32, 38], writing style [27], and combination statistical features [22, 41]. Recently, the focus gradually shifted towards end-to-end approaches without explicitly relying on specific features [43, 48]. Notably, utilizing Graph Neural Network (GNN) to capture structural relationships gained popularity [4, 39]. These approaches mainly include modeling users' relationship [5, 11], modeling the relationship between topics and comments [3, 30, 54], and examining controversies through introducing entities and polarity [39], and so on. In addition, researchers have also explored early comments to predict controversy [23]. More recently, with advancements in Pretrained Language Models, they have gradually been employed in controversy detection and other related tasks [7, 9, 46]. However, the aforementioned methods for controversy detection are mostly limited to textual modality. To the best of our knowledge, there have been no approaches developed for multimodal controversy detection thus far.

---

[1]Upon acceptance of this paper, the codebase will be made publicly available at https://anonymous.4open.science/r/MM_Controversy_Detection_Released-DE6A.
[2]https://www.theguardian.com
[3]https://www.emol.com/
[4]https://www.toutiao.com

 

**Table 1: Summary of controversy detection datasets. The term "metadata" refers to fundamental statistical indicators including the number of likes, forwards, and comments.**

| Dataset | Feature | | | | | Category | Language | Accessibility | Source | Time Span |
|---------|---------|------|----------|---------|---------|----------|----------|---------------|--------|-----------|
|         | Video | Text | Metadata | Comment | Profile |          |          |               |        |           |
| Dori et al. [17] | - | ✓ | - | - | - | Website | English | NOT-public | Wikipedia | - |
| Beelen et al. [2] | - | ✓ | - | - | - | News | English | NOT-public | theguardian.com | 2017.09-2017.11 |
| Twitter Pages [11] | - | ✓ | - | - | ✓ | Social media | English | NOT-public | Twitter | - |
| Linmans et al. [31] | - | ✓ | - | - | - | Website | English | NOT-public | Wikipedia | 2018-2019 |
| Hessel et al. [23] | - | ✓ | - | ✓ | ✓ | News | English | NOT-public | Reddit | 2007.01-2014.02 |
| Zhong et al. [54] | - | ✓ | - | ✓ | - | Social media | Chinese | Partly-public | Weibo | 2017.07-2019.08 |
| Mendoza et al. [39] | - | ✓ | ✓ | - | ✓ | News | English | NOT-public | emol.com | 2016.04-2019.04 |
| De França et al. [13] | - | ✓ | - | - | - | Social media | English | Partly-public | Twitter | 2021.02-2021.04 |
| Canute et al. [9] | - | ✓ | - | - | - | Social media | English | All-public | Twitter | 2020.01-2022.12 |
| Li et al. [30] | - | ✓ | - | - | - | News | Chinese | All-public | Toutiao | 2019.03-2019.12 |
| ProsCons [48] | - | ✓ | - | ✓ | - | Social media | Chinese | NOT-public | Weibo | 2021.03-2022.03 |
| **MMCD (ours)** | ✓ | ✓ | ✓ | ✓ | ✓ | **Social media** | **Chinese** | **All-public** | **Douyin** | **2017.12-2023.12** |

## 3 THE MMCD DATASET

To fill the existing void of publicly available datasets, we introduce the Multimodal Controversy Dataset (MMCD). This dataset comprises over 10,000 Chinese videos, each accompanied by a wealth of social context information. Our intention in creating this dataset is to offer researchers an invaluable resource for studying multimodal controversy detection. By providing MMCD, we enable the development and evaluation of innovative approaches in this area of research.

### 3.1 Data Collection

We collected raw videos from Douyin[5], a popular Chinese social video platform, known for its vast user base of millions of active participants. To obtain a comprehensive set of controversial videos and establish reliable ground truth labels for controversy, we manually formed a set of 139 keywords (listed in supplementary material) based on the popular news rankings on Weibo. Leveraging these query keywords, we searched for videos and crawled relevant content.

Our data collection primarily involves crawling for *video content*, *metadata*, *publishers' profiles*, and *comments context*. The video content category encompasses fundamental attributes including videos' IDs, publication timestamps, descriptions, URLs, and lengths; metadata includes metrics including the number of likes, shares, and comments; publishers' profile contains relevant information about the publisher, providing insights into their characteristics. Additionally, the comment category provides details regarding user comments associated with the videos. Notably, the comments are sorted by the Douyin platform, mostly considering factors such as the comments' popularity and timestamp, and we selected the Top 40 from them. To ensure data quality, we implemented a filtering process to exclude aberrant unplayable videos. As a result, we collected a total of 18,623 Chinese videos released between Dec. 2017 and Dec. 2023.

### 3.2 Data Annotation

We implemented a meticulous manual annotation process. During the selection of annotators, we prioritize maximizing demographic diversity and including individuals from various cultural backgrounds. Our group of 25 annotators consists of 2 Ph.D. students, 3 graduate students, and 20 undergraduate students from 5 departments at our university. Among the annotators, 11 identified as women and 14 as men, with ages ranging from 18 to 30.

Annotating multimodal data presents additional challenges compared to annotating textual data, primarily due to the requirement for annotators to watch lengthy videos. To address this challenge and ensure consistent annotation quality across all videos, annotators were provided with explicit instructions through a comprehensive guideline prepared by us. These guidelines instruct annotators to evaluate the level of controversy in three aspects: (1) the controversy within the video itself, (2) the controversy between the video and its comments, and (3) the controversy among the comments. Based on these assessments, annotators classified each video as either controversial or non-controversial. We developed a website for the annotation process. To ensure reliability, each video was annotated by three annotators. The annotation process resulted in substantial consensus among the annotators, with a Kappa value of 0.78 indicating significant agreement. Following the annotation process, we obtained a dataset consisting of 5,643 controversial videos and 11,164 non-controversial videos.

### 3.3 Data Analylsis

To gain insights into the distinctive characteristics of controversial and non-controversial videos, we conducted an exploratory analysis of the collected dataset from three perspectives: data distribution, indicators statistics, and sentiment analysis. These analyses aim to provide valuable insights into the underlying behaviors and patterns associated with these two types of videos, thereby contributing to controversy detection.

**Data Distribution.** The MMCD Dataset is categorized into 14 domains using DBpedia [1] and manual judgment (details in supplementary materials). Figure 2 illustrates the distribution of data across these different domains. It is observed that different domains exhibited varying levels of controversy, resulting in an imbalanced

---
[5]https://www.douyin.com

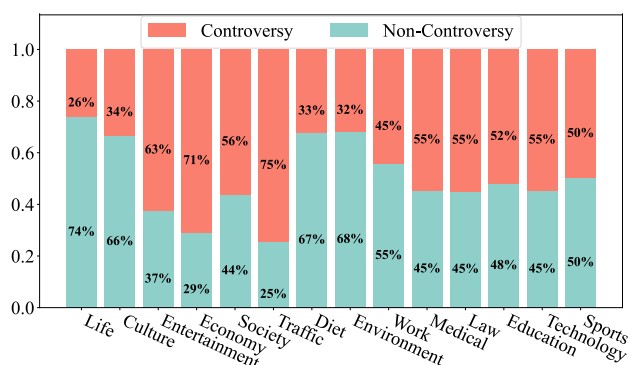

Figure 2: Data distributions across different domains.

Table 2: Statistical analysis of videos on MMCD.

| Data Type | Contro. Video | Non-Contro. Video |
|---|---|---|
| Length of video | 87s | 79s |
| #(Forwards) | 8,480 | 5,245 |
| #(Likes) | 143,300 | 83,300 |
| #(Comments) | 13,900 | 12,700 |
| #(Publishers' Videos) | 4,960 | 3,467 |
| #(Publishers' Likes) | 162,450,000 | 141,170,000 |
| #(Publishers' Followers) | 3,038,317 | 2,757,106 |

dataset. Among these domains, the "Life" exhibits the least degree of controversy in its data distribution. This can be attributed to the close connection between "Life" and everyday experiences, which enables individuals to have a deeper understanding of this domain. On the other hand, the "Traffic" domain exhibits the highest degree of controversy in its data distribution. This can be primarily attributed to the complex nature of traffic conditions, making it difficult to determine responsibility for accidents. Moreover, conflicts arising from conflicting interests among different parties further contribute to the elevated controversy surrounding this domain.

**Indicators Statistics.** We extensively analyzed various video content indicators, as presented in Table 2. These indicators include the average length of videos, the number of forwards, likes and comments, as well as the number of videos & likes associated with publishers. Based on the statistical findings, we observed the following patterns. First, regarding video length, it was observed that controversial videos tend to be longer on average. This suggests that controversial videos likely require more substantial content to stimulate user discussions. Second, it was noted that controversial videos garnered a higher number of forwards, likes, and comments compared to non-controversial videos. This implies that controversial videos have a greater propensity to attract attention and engagement from viewers. Additionally, publishers of controversial videos tend to have more videos, likes, and followers, which implies that individuals who disseminate controversial content tend to be more active.

Figure 3 depicts the distribution of video counts and corresponding likes by publisher. An intriguing observation arises from the graph, where the distribution of likes and video counts for publishers of controversial videos exhibits a higher degree of scattering,

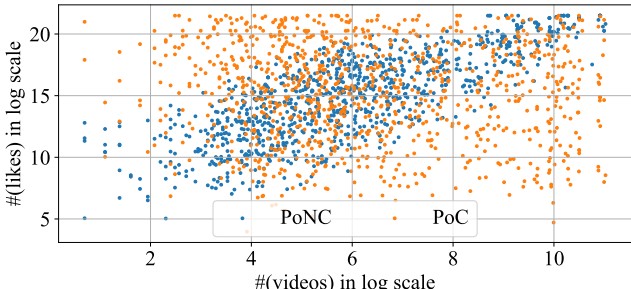

Figure 3: Comparison of likes and video counts among publishers. PoC refers to publishers of controversial videos, while PoNC refers to publishers of non-controversial videos.

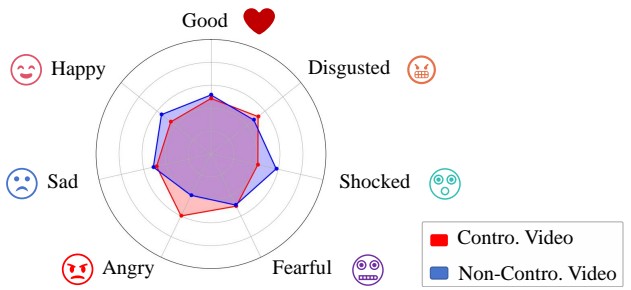

Figure 4: Comparison of fine-grained emotions between controversial and non-controversial videos.

suggesting diverse behavior among them. This observation may suggest two potential explanations: 1) publishers with a low video count but a high number of likes may represent specialized marketing accounts that attract viewers with sensational content; 2) publishers with a high video count but a low number of likes could potentially be engaged in bot-like behavior, actively spreading controversial videos.

**Sentiment Analysis.** We conducted sentiment analysis on the textual data present in the crawled dataset, including video descriptions, comments, and ASR text. To perform a fine-grained analysis, we utilized the CNsenti tool [51], which covers seven emotion categories: good, happy, sad, angry, fearful, disgusted, and shocked. Figure 4 illustrates the average scores of these fine-grained emotions. Upon comparison, we observed that controversial videos exhibited higher scores for emotions like "angry" and "disgusted", while non-controversial videos received higher scores for the emotions of "good", "happy", and "shocked".

## 4 METHOD

We propose the Multi-view Controversy Detection (MVCD) framework, which integrates various modules for detecting controversial videos. Figure 5 illustrates the architecture of MVCD, comprising five components: (a) *Multimodal Feature Extraction*, which extracts multimodal features by pre-trained models; (b) *Modality Awareness Learning (MAL)*, which integrates multiple features to learn the overall controversial features of video content; (c) *Contextual Graph Learning (CGL)*, which models the relationship between videos and comments. (d) *Inconsistency Enhanced Learning (IEL)*, which focuses

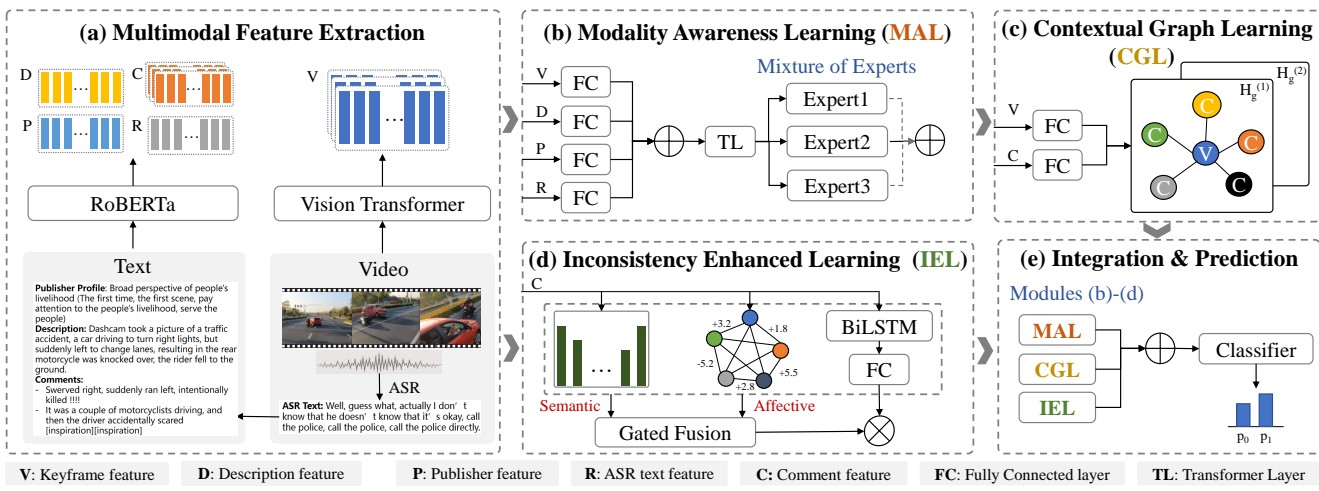

**Figure 5: Architecture of Multi-view Controversy Detection (MVCD) framework.**

on capturing the controversial inconsistency of comments; and (e) *Integration & Prediction*, to concatenate the features generated by the (b)-(d) modules and classify whether the input is controversial.

### 4.1 Problem Formulation

MVCD aims to detect whether a given video and its associated content is controversial. Formally, let $\mathcal{T}$ denote an input sample consisting of six elements: video keyframes $\mathcal{V}$, description $\mathcal{D}$, comments $\mathcal{C}$, publisher profiles $\mathcal{P}$, ASR text $\mathcal{R}$, and ground truth label $y$. The length of these elements are represenes as $n_d$, $n_c$, $n_p$, and $n_r$, respectively. Our objective is to develop a multi-modal controversy detection model F:

$$\hat{y} = F\left(\mathcal{V}, \mathcal{D}, \mathcal{C}, \mathcal{P}, \mathcal{R} \mid \Theta\right), \quad (1)$$

where $\hat{y}$ denotes the binary classification prediction result for a given sample obtained by model F, and $\Theta$ represents the set of all parameters associated with the model.

### 4.2 Multimodal Feature Extraction

To effectively capture the diverse characteristics present in various modalities, we leverage a pretrained Chinese model CN-CLIP [52], which is trained on large-scale image-text pairs.

**Video.** To extract frame-level features, we first utilize the ffmpeg tool[6] to extract keyframes from each video. These keyframes are then passed through the image encoder. The resulting encoded video representation is denoted as $\mathbf{X}^v = [\mathbf{x}_1^v, ..., \mathbf{x}_{n_v}^v]$, where $\mathbf{x}_i^v \in \mathbb{R}^{d_v}$ represents the feature vector extracted from the $i$-th keyframe, $n_v$ signifies the total number of keyframes in the video, and $d_v$ denotes the dimension of the image encoded by Vision Transformer (ViT) [19] in CN-CLIP.

**Text.** We employ RoBERTa [33] for text feature extraction in CN-CLIP. The features extracted from the text elements (description, publisher's profile, ASR text) are represented as $\mathbf{x}^d$, $\mathbf{x}^p$, $\mathbf{x}^r \in \mathbb{R}^{d_t}$, where $d_t$ represents the dimension of the encoded text sequences. Specifically, for publisher profiles, we concatenate attributes such

[6]https://git.ffmpeg.org/ffmpeg.git

as nickname, personal introduction, verification information, the number of videos and likes. As for comments, The features are represented as $\mathbf{X}^c = [\mathbf{x}_1^c, ..., \mathbf{x}_{n_c}^c]$, where $\mathbf{x}_i^c \in \mathbb{R}^{d_t}$ represents the feature vector of the $i$-th comment within $\mathbf{X}^c$, and $n_c$ denotes the total number of comments.

### 4.3 Modality Awareness Learning

Features extracted from video keyframes, descriptions, publishers' profiles, and ASR text are passed through individual Fully Connected layers (FC) to align multimodal features. These aligned features are then concatenated and input into a single Transformer Layer (TL) to capture temporal information. The calculations involved in this process are as follows:

$$\mathbf{H}_{mod} = \sigma\left(\mathbf{W}_{mod}\mathbf{X}^{mod} + \mathbf{b}_{mod}\right), mod \in \{v, d, p, r\}, \quad (2)$$

$$\mathbf{H}_o = TL\left(\text{Concat}\left([\mathbf{H}_v, \mathbf{H}_d, \mathbf{H}_p, \mathbf{H}_r]\right)\right), \quad (3)$$

where $\mathbf{W}_v$, $\mathbf{W}_d$, $\mathbf{W}_p$, and $\mathbf{W}_r$ denote the weight parameters, while $\mathbf{b}_v$, $\mathbf{b}_d$, $\mathbf{b}_p$, and $\mathbf{b}_r$ represent the corresponding bias parameters.

Considering the diverse impacts of different modalities on the perception of varied audiences, it is imperative to tackle the challenges arising from inconsistent and insufficient attention given to these modalities. To overcome these challenges, we utilize the Mixture of Expert (MoE) architecture [44] to enhance the overall modeling performance. The MoE layer consists of a set of $m$ expert networks denoted as $E(\cdot)$, along with a gating network referred to as $G(\cdot)$. The output feature $\mathbf{H}_o$ from the previous step is fed into the gating network $G(\cdot)$ and expert network $E(\cdot)$ to gain the output:

$$G(\mathbf{H}_o) = \text{Softmax}\left(\text{KeepTopK}(\mathbf{W}_{moe}\mathbf{H}_o, k)\right), \quad (4)$$

$$\mathbf{Z}_1 = \sum_{q=1}^{m} G(\mathbf{H_o})_q E_q(\mathbf{H_o}), \quad (5)$$

where $\mathbf{W}_{moe}$ denotes learnable parameters, KeepTopK [44] is a function to select Top $k$ highest gate values given input feature $\mathbf{H}_o$, and $q$ denotes the ordinal position of the expert network.

## 4.4 Contextual Graph Learning

To effectively capture the controversy present between videos and comments, we employ the Graph Convolutional Network (GCN) [25] to capture the semantic and structural relationships that exist between these two modalities. The comments feature $\mathbf{X}^c$ is aligned with $\mathbf{X}^v$ to establish a unified feature:

$$\mathbf{H}_c = \sigma\left(\mathbf{W}_c \mathbf{X}^c + \mathbf{b}_c\right), \tag{6}$$

where $\mathbf{W}_c$ denotes the weight parameter, and $\mathbf{b}_c$ represents the bias parameter. We then construct a Video-Context graph denoted as $\mathcal{G} = (\mathcal{V}, \mathcal{E})$ for each video, where the set of nodes $\mathcal{V}$ consist of features of video or comment, and an edge exists between a video and its corresponding comments when the comment is associated with that specific video.

The initial representations of the nodes can be defined as:

$$\mathbf{V}^{(0)} = [\mathbf{v}_1, ..., \mathbf{v}_{1+n_c}] = [\mathbf{H}_v, \mathbf{H}_c]. \tag{7}$$

During the message-passing process, each node updates its representation based on the aggregated information obtained from its neighboring nodes and its features. This allows the learned representation to encompass valuable insights from both the content and structure of the graph. Specifically, for a given node $\mathbf{v}_i \in \mathbf{V}^{(0)}$, the update rule can be expressed as:

$$\mathbf{v}_i^{(l+1)} = \sigma\left(\sum_{j \in \mathcal{N}_i} g\left(\mathbf{v}_i^{(l)}, \mathbf{v}_j^{(l)}\right) + \mathbf{b}^{(l)}\right), \tag{8}$$

where $\mathbf{v}_i^{(l)}$ represents the hidden state of node $v_i$ in the $l$-th layer of GCN, $\sigma$ denotes the Rectified Linear Unit (ReLU) activation function, $\mathcal{N}_i$ denotes the neighbors of node $v_i$ (including the node itself), $g(\cdot)$ is the aggregation function, and $\mathbf{b}^{(l)}$ represents the bias term.

At the layer level, we use the embedding vectors $\mathbf{V}^{(0)}$ as input to a two-layer GCN, resulting in a condensed representation denoted as $\mathbf{V}^{(2)}$. Incoming messages from the neighbor set $\mathcal{N}_i$ are aggregated by $g(\cdot)$, which is implemented as a linear function. Thus, for the $l$-th layer, the propagation rule is given by:

$$\mathbf{V}^{(l+1)} = \sigma\left(\hat{\mathbf{A}} \mathbf{V}^{(l)} \mathbf{W}^{(l)} + \mathbf{b}^{(l)}\right), \tag{9}$$

where $\mathbf{V}^{(l)}$ contains all node vectors in the $l$-th layer, $\hat{\mathbf{A}}$ is the normalized adjacency matrix, $\mathbf{W}^{(l)}$ is the weight matrix.

At last, we employ MaxPooling to extract the most significant features for subsequent calculations:

$$\mathbf{Z}_2 = \text{MaxPooling}\left(\mathbf{V}^{(2)}\right). \tag{10}$$

## 4.5 Inconsistency Enhanced Learning

Given the intrinsic inconsistencies observed in controversial comments, which often include discrepancies in content and sentiment, we propose utilizing an affective matrix and semantic attention matrix [35] to capture and model these inconsistencies.

**Context Affective Computing.** To capture and analyze the affective inconsistencies in comments, we construct an affective matrix $\mathbf{A} \in R^{n_c * n_c}$ based on a set of comments $C = \{c_1, c_2, ..., c_{n_c}\}$. Each element $a_{i,j}$ in $\mathbf{A}$ is computed by:

$$a_{i,j} = \left|u\left(c_i\right) - u\left(c_j\right)\right|, \tag{11}$$

where $u(c_i)$ denotes the affective score of comment $c_i$ calculated using an external sentiment dictionary SenticNet [8], and $|\cdot|$ represents the absolute value calculation.

By employing this approach, the weight of the corresponding edge increases in proportion to the magnitude of sentiment reversal between every two comments, allowing significant attention to be directed towards comments that exhibit opposing sentiments.

**Context Semantic Computing.** We compute the semantic attention matrix $\mathbf{T}$ to measure the semantic inconsistency between comments. Specifically, for each pair of comment features $(\mathbf{x}_i^c, \mathbf{x}_j^c)$, $i, j \in (1, 2, ..., n_c)$, the attention score $t_{i,j}$ is calculated as:

$$t_{i,j} = \sigma(\mathbf{x}_i^c \mathbf{W}_{i,j})\sigma(\mathbf{x}_j^c \mathbf{W}_{i,j})^T, \tag{12}$$

where $\mathbf{W}_{i,j}$ is a trainable parameter matrix, and $(\cdot)^T$ signifies matrix transposition.

**Fusion & Enhanced.** We incorporate the learned affective features $\mathbf{A}$ and semantic attention features $\mathbf{T}$ for a more expressive representation:

$$\mathbf{T}_a = \alpha \mathbf{A} + (1 - \alpha)\mathbf{T}, \tag{13}$$

where $\alpha \in \mathbb{R}$ is the hyperparameter. In order to comprehensively analyze the comment features $\mathbf{X}^c = [\mathbf{x}_1^c, ..., \mathbf{x}_{n_c}^c]$ we employ a Bidirectional Long Short-Term Memory (BiLSTM) model to capture contextual information $\mathbf{C} = [\mathbf{c}_1, ..., \mathbf{c}_{n_c}]$:

$$\mathbf{C} = \text{BiLSTM}\left(\mathbf{X}^c\right). \tag{14}$$

To simultaneously consider these two inconsistencies, we employ the inner product operation to obtain the output:

$$\mathbf{O} = \mathbf{T}_a(\mathbf{C}\mathbf{W}_a), \tag{15}$$

where $\mathbf{W}_a$ is the weight matrix. Subsequently, the MaxPooling function is applied to derive the final output $\mathbf{Z}_3$:

$$\mathbf{Z}_3 = \text{MaxPooling}\left(\mathbf{O}\right). \tag{16}$$

## 4.6 Integration & Prediction

To obtain the final integration outputs from the three modules, we concatenate $\mathbf{Z}_1, \mathbf{Z}_2, \mathbf{Z}_3$, and pass them through a constructed classifier to obtain the final outputs. The classifier consists of a stacked architecture with two fully connected layers comprising layer normalization, ReLU, and dropout. The final probability distributions are calculated by:

$$\mathbf{Z}_o = \text{Concat}\left([\mathbf{Z}_1, \mathbf{Z}_2, \mathbf{Z}_3]\right), \tag{17}$$

$$\mathbf{p} = \sigma\left(\text{LN}\left(\mathbf{W}_o' \mathbf{Z}_o + \mathbf{b}_o'\right)\right)\mathbf{W}_o + \mathbf{b}_o, \tag{18}$$

where $\mathbf{W}_o$, $\mathbf{W}_o'$, $\mathbf{b}_o$ and $\mathbf{b}_o'$ are model parameters, and LN means Layer Normalization function. The probability matrix $\mathbf{p}$ comprises $p_0$ and $p_1$, representing the predicted probability for the label being 0 (non-controversy) and 1 (controversy), respectively. Ultimately, the predicted label $\hat{y}$ is defined as:

$$\hat{y} = \text{argmax}\left([p_0, p_1]\right). \tag{19}$$

To train the whole framework, we combine three loss functions. First, we use the cross-entropy loss function to measure dissimilarity between predicted probabilities and ground truth labels. Given that $y \in \{0, 1\}$ denotes the ground truth label, the loss function is calculated as:

$$\mathcal{L}_{ce} = -[(1 - y)\log p_0 + y \log p_1]. \tag{20}$$

Additionally, we employ a balancing loss for the MoE layer to ensure fair load and importance among experts. The calculation is as follows:

$$\mathcal{L}_{moe} = w_{imp} \, \text{CV} \, (\text{G}(\mathbf{H}_o))^2 + w_{ld} \, \text{CV} \, (\text{P}(\mathbf{H}_o))^2, \quad (21)$$

where CV denotes the coefficient of variation, $\text{P}(\cdot)$ is the smooth function described by [44], and the hyperparameters $w_{imp}$ and $w_{ld}$ are used to balance expert importance and load. Furthermore, we add a regularization loss to improve the quality of learned semantic information:

$$\mathcal{L}_{reg} = w_{spa} \, \|\mathbf{T}\|_F^2, \quad (22)$$

where $\| \cdot \|_F$ is the Frobenius norm of a matrix, and $w_{spa}$ is the sparsity hyperparameter.

Finally, we sum up the three loss functions to obtain the ultimate loss function:

$$\mathcal{L} = \mathcal{L}_{ce} + \mathcal{L}_{moe} + \mathcal{L}_{reg}. \quad (23)$$

## 5 EXPERIMENTS

We conducted experiments to address the following research questions:

- **RQ1:** Is our proposed MVCD framework more effective than traditional and state-of-the-art baselines?
- **RQ2:** Does our proposed model effectively utilize multimodal information, and do the individual modules within the model provide substantial contributions?
- **RQ3:** Does the model demonstrate effectiveness in scenarios with limited availability of comments?

## 5.1 Baselines

We establish a comprehensive benchmark for controversy detection by conducting experiments using multiple representative methods as baselines, including uni-modal and multi-modal models.

**Uni-modal.** Due to the predominant focus of previous research on textual modalities, our experiments include several uni-modal baselines, particularly those based on text modality. We utilize BERT [15] and RoBERTa [33] as representative baselines for pre-trained language models. For the pretrained image model, we employ ViT [19]. In addition, we incorporate representative models such as TPC-GCN [54] and DTPC-GCN [54] as baselines in the controversy detection task. Furthermore, we use foundation models known for their state-of-the-art capabilities across various downstream tasks, including ChatGLM3-6B [20] and GPT3.5 [6].

**Multi-modal.** Since multimodal approaches are not yet widely used in controversy detection, we select related multimodal tasks such as fake detection and social bot detection. For the fake detection task, we consider two representative models: MCAN [50] and SVFEND [42]. For the social bot detection task, we include Bot-MoE [34] as a baseline model. Furthermore, we employ state-of-the-art foundation models including VideoChat [29], ChatGLM4 [20] and GPT4 [6] for comparison.

## 5.2 Experimental Settings

**Data Preprocessing.** To ensure sample balance, we select an equal number of controversial and non-controversial videos from the annotated dataset, resulting in 5,632 videos each. The dataset is divided into training, validation, and test sets using an 8:1:1 ratio. Data preprocessing includes removing hashtags from video descriptions and eliminating stopwords such as mentions (@person) from comments. As for audio, we utilize the Baidu API[7] to perform Automatic Speech Recognition to obtain text.

**Implementation Details.** In our experiments, we employ CN-CLIP [52] to generate image and text embeddings with a fixed vector size of 1024. The Adam [24] optimizer is used to optimize parameters with a learning rate set to 1e-4. The classification dimension is set to 128, and the batch size is set to 128. Training extends for 100 epochs, incorporating early stopping if the validation score does not improve for 10 consecutive epochs. For VideoChat, we employ the gpt-3.5-turbo-16k model for language generation, setting the frame sampling frequency parameter to 4. In the case of GPT3.5 and GPT4V, the temperature is set to 0.7. More details regarding the prompts used in the foundation models are presented in the supplementary materials.

## 5.3 Experimental Results (RQ1)

Table 3 shows the quantitative results of the evaluation. The experiment results indicate the following observations: MVCD demonstrates superior performance compared to other approaches, validating its effectiveness in capturing important multimodal clues for detecting controversial videos.

Surprisingly, the performance of foundation models (GPT3.5, VideoChat, ChatGLM4, and GPT4V) is not ideal, both in uni-modal and multimodal scenarios. This could be attributed to the lack of appropriate training datasets for multimodal controversy detection, which may prevent large models from fully learning diverse controversial scenarios.

Regarding SVFEND, which incorporates audio data during the training process, a notable performance disparity is observed when compared to all baselines, except for the foundation models. Remarkably, we observed that the training accuracy of SVFEND achieves an exceptional value of 0.99, without a corresponding proportional increase in testing accuracy. Once we eliminated the audio features from the model, we observed a more reasonable trend in training accuracy, accompanied by an improvement in testing. These findings lead us to speculate that including the audio modality during training may result in overfitting, leading to suboptimal performance in the outcomes.

## 5.4 Ablation Study (RQ2)

We conduct a series of ablation experiments to evaluate the importance of each modality and module in detecting controversial videos. The results, as shown in Table 4, indicate that all modules perform well in the multimodal controversy detection task. Notably, when combined as a full model (MVCD), it achieves the highest accuracy of 72.62%. The CGL module exhibits slightly superior performance compared to the others, highlighting the importance of modeling the relationships between videos and comments. Furthermore, considering the involvement of multiple modalities in multimodal controversy detection, we performed ablation experiments on various data features. The experimental results indicate that each data feature plays a role, with notable contributions from profile and video features.

---

[7]https://vop.baidu.com/server_api

**Table 3: Performance (%) comparison among different methods on MMCD in terms of F1-score, recall, precision, and accuracy.**

| Modality | Video | Audio | Text | | | | | Method | F1-score | Recall | Precision | Accuracy |
|---|---|---|---|---|---|---|---|---|---|---|---|---|
| | | | D† | C | P | R | K | | | | | |
| Uni-modal | - | - | ✓ | ✓ | ✓ | - | - | BERT [15] | 61.57 | 63.96 | 68.57 | 63.96 |
| | - | - | ✓ | ✓ | ✓ | - | - | RoBerta [33] | 65.54 | 66.52 | 68.64 | 66.52 |
| | ✓ | - | - | - | - | - | - | ViT [19] | 64.34 | 64.84 | 65.72 | 64.84 |
| | - | - | ✓ | ✓ | - | - | ✓ | TPC-GCN [54] | 66.84 | 67.58 | 69.30 | 67.58 |
| | - | - | ✓ | ✓ | - | - | ✓ | DTPC-GCN [54] | 67.14 | 67.14 | 67.14 | 67.14 |
| | - | - | ✓ | ✓ | ✓ | ✓ | - | ChatGLM3 [20] | 44.22 | 48.46 | 47.67 | 49.66 |
| | - | - | ✓ | ✓ | ✓ | - | - | GPT3.5 [6] | 36.86 | 50.44 | 53.16 | 50.44 |
| Multi-modal | ✓ | - | ✓ | ✓ | ✓ | - | - | MCAN [50] | 65.09 | 65.46 | 66.15 | 65.46 |
| | ✓ | - | ✓ | ✓ | ✓ | - | - | BotMOE [34] | 67.27 | 67.31 | 67.40 | 67.31 |
| | ✓ | ✓ | ✓ | ✓ | ✓ | - | - | SVFEND [42] | 57.33 | 58.66 | 59.89 | 58.66 |
| | ✓ | - | ✓ | ✓ | ✓ | ✓ | - | VideoChat [29] | 47.58 | 50.09 | 50.11 | 50.09 |
| | ✓ | - | ✓ | ✓ | ✓ | ✓ | - | ChatGLM4 [20] | 38.15 | 49.51 | 48.39 | 48.04 |
| | ✓ | - | ✓ | ✓ | ✓ | ✓ | - | GPT4V [6] | 58.10 | 61.06 | 65.10 | 60.79 |
| | ✓ | - | ✓ | ✓ | ✓ | ✓ | - | **MVCD (ours)** | **72.46** | **72.62** | **73.15** | **72.62** |

† Data formats are abbreviated for simplicity (D: Description, C: Comment, P: Profile, R: ASR Text, K: Keyword).

**Table 4: Experimental results of ablation study.**

| Category | Method | F1 | Rec. | Prec. | Acc. |
|---|---|---|---|---|---|
| Feature | w/o V† | 68.91 | 68.99 | 69.19 | 68.99 |
| | w/o D | 71.46 | 71.47 | 71.48 | 71.47 |
| | w/o C | 69.75 | 69.79 | 69.89 | 69.79 |
| | w/o P | 68.04 | 68.20 | 68.57 | 68.20 |
| | w/o R | 70.84 | 71.03 | 71.57 | 71.03 |
| Module | MAL-only | 69.75 | 69.79 | 69.89 | 69.79 |
| | CGL only | 69.94 | 70.14 | 70.69 | 70.14 |
| | IEL only | 69.16 | 69.17 | 69.20 | 69.17 |
| | w/o MoE | 70.03 | 70.05 | 70.12 | 70.05 |
| Full Model | **MVCD (ours)** | **72.46** | **72.62** | **73.15** | **72.62** |

† Data formats are abbreviated for simplicity (D: Description, C: Comment, P: Profile, R: ASR Text, V: Video keyframe).

**Table 5: Experimental results of early prediction.**

| Time | Method | F1-score | Recall | Precision | Accuracy |
|---|---|---|---|---|---|
| 0h | MVCD | **67.91** | **67.93** | **67.99** | **67.99** |
| | CGL & IEL | 50.49 | 50.62 | 50.63 | 50.62 |
| <1h | MVCD | **67.71** | **67.76** | **67.86** | **67.76** |
| | CGL & IEL | 50.75 | 50.88 | 50.89 | 50.88 |
| <2h | MVCD | **68.46** | **68.46** | **68.46** | **68.46** |
| | CGL & IEL | 66.52 | 66.61 | 66.78 | 66.61 |
| <3h | MVCD | **68.29** | **68.29** | **68.29** | **68.29** |
| | CGL & IEL | 65.69 | 65.72 | 65.78 | 65.72 |
| <4h | MVCD | **69.24** | **69.26** | **69.31** | **69.26** |
| | CGL & IEL | 64.79 | 64.84 | 64.92 | 64.84 |
| <5h | MVCD | **70.41** | **70.41** | **70.41** | **70.41** |
| | CGL & IEL | 65.16 | 65.19 | 65.26 | 65.19 |

## 5.5 Early Prediction (RQ3)

During the initial stages of video publication, despite limited interaction and comments, detecting controversies is crucial for enhancing content quality and fostering audience engagement. To assess the model's performance under such circumstances, we evaluate its effectiveness using comments posted within 5 hours after the video is released. We compare two models: the joint model (CGL & IEL) that incorporates Contextual Graph Learning (CGL) and Inconsistency Enhanced Learning (IEL) with comments playing a pivotal role, and the full model MVCD. The experimental results are presented in Table 5. The joint model (CGL & IEL) achieves relatively low accuracy rates of 50.62% and 50.88% within 0 and 1 hour, respectively. However, the full model MVCD, aided by the VGI module that effectively utilizes video content for prediction, yields more effective results and mitigates the significant drop in performance. This suggests that multimodal content plays a significant role in situations where comments are limited.

## 6 CONCLUSION AND OUTLOOKS

In this work, we released a comprehensive Multimodal Controversy Dataset (MMCD) in Chinese. To gain a thorough understanding of the characteristics exhibited by controversial videos, we have conducted an exploratory analysis of MMCD. Additionally, to facilitate further research in this field, we have proposed a Multi-view Controversy Detection framework, which effectively captures controversies presented within the video content itself, as well as those arising from the interaction between the video and its associated comments, or among the comments themselves. Extensive experiments showed the effectiveness of the proposed framework.

Future research should explore integrating summarized viewpoints in videos or comments for valuable controversy detection. Detecting support and opposition in viewpoints is crucial for controversy detection. Emphasizing ethical implications and explainability of model predictions is essential, especially in sensitive contexts.

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
