# OpenReview forum: "A Chinese Multimodal Social Video Dataset for Controversy Detection"
_acmmm.org/ACMMM/2024/Conference — MM2024 Poster_

### Official Review · Reviewer_38QG · 2024-05-10

**Rating:** 5
**Confidence:** 2

**Summary:**

This paper introduces a novel multimodal social video dataset MMCD for controversy detection, and a multimodal framework, named MVCD, for detecting controversaries.

**Strengths:**

1.	The introduced MMCD dataset is novel, and is expected to advance this research field.
2.	The proposed MVCD outperforms the baselines by a large margin.

**Limitations:**

1.	In the Introduction, I would recommend to brief the components of MVCD.
2.	In Section 3.2, it seems that all the annotators are students from the same university, this may potentially introduce biases in the annotated label.
3.	In the baselines, are MCAN and SVFEND designed to process Chinese? If they are not designed to process Chinese, do you modify their components to better process Chinese. If not, the experiments may not be fair, because the proposed method leverages a pretrained Chinese model CN-CLIP to process Chinese text.

**Suitability:**

3

---

### Official Review · Reviewer_YhXs · 2024-05-19

**Rating:** 5
**Confidence:** 3

**Summary:**

This paper introduces the MMCD, a Multimodal Controversy Dataset in Chinese, and provides a thorough analysis of this newly constructed dataset. The authors also propose a Multi-View Controversy Detection(MVCD) framework that captures controversies presented within the video content itself, as well as those arising from the interaction between the video and its associated comments or among the comments themselves. Extensive experiments demonstrate the effectiveness of the proposed framework.

**Strengths:**

1. **New Dataset:** The authors constructed a large-scale Chinese multimodal controversy dataset, MMCD. If made publicly available, this dataset would significantly contribute to the academic community's research on controversy detection in short videos, particularly within the Chinese context.
2. **Well-Written Manuscript:** The paper is well-organized and clearly written, making it easy for readers to follow.
3. **Rich Dataset Analysis:** The authors provide a comprehensive analysis of their dataset, offering valuable insights and findings that can inform and inspire further research in controversy detection.
4. **Meaningful Experiments:** The paper includes extensive experiments that effectively demonstrate the proposed framework's performance. Each experiment contributes to a deeper understanding of the model's capabilities and the importance of its components.

**Limitations:**

1. **Data Analysis Enhancement:** The data analysis could be further improved. For instance, providing a quantitative breakdown of the number of videos across different domains before analyzing the distribution of controversy data would give readers a clearer context.
2. **Ethical Concerns:** The model uses nicknames directly, which raises potential ethical issues. The authors should consider addressing these concerns and discuss how they ensure the privacy and ethical handling of data.
3. **Code Availability:** Although the authors have provided an anonymous GitHub link, they did not include their code upon submission. Making the code available would greatly benefit the research community by allowing others to replicate and build upon their work.
4. **Case Studies:** Including case studies would strengthen the paper. Specific examples showing how different components like MAL, CGL, and IEL, contribute to the final controversy detection would provide practical insights into the model's application and effectiveness.

**Suitability:**

3

---

### Official Review · Reviewer_SQrB · 2024-05-23

**Rating:** 5
**Confidence:** 3

**Summary:**

This work constructs a large-scale Multimodal Controversial Dataset (MMCD) in Chinese and proposes a novel framework named Multi-view Controversy Detection (MVCD) to effectively model controversies. The collected dataset from Douyin, a Chinese video platform, has been manually annotated, with annotators assessing the controversy within the video itself, the controversy between the video and its comments, and the controversy among the comments. The videos are diverse, covering 14 different domains. Additionally, the authors provided a statistical analysis of the videos and their popularity in the dataset, alongside sentiment analysis. This analysis provides a good overview of the MMCD dataset.

Regarding the detection approach, the authors propose a complex architecture consisting of five components. Three of the components capture the different aspects of controversy (within the video, between the video and comments, and among comments), while the final component integrates the features of the previous components and predicts whether the video is controversial. The choices for each component are quite reasonable and state-of-the-art, ranging from Mixture of Experts (MoE) to Graph Convolutional Networks. The baselines for comparison are both unimodal and multimodal, and the proposed architecture outperforms all of them. The ablation study is thorough and provides useful insights, as the complexity of the proposed architecture makes it necessary to understand and justify each part. Finally, the third research question for early controversy prediction is quite interesting, proving the potential for real-world applications.

**Strengths:**

1) The dataset from Douyin is diverse, covering 14 domains, and has been manually annotated for various aspects of controversy.

2) The proposed architecture outperforms both unimodal and multimodal baselines.

3) The ablation study is thorough and provides useful insights into the architecture's effectiveness

**Limitations:**

The complexity of the architecture makes it difficult to understand the overall architecture, and despite the ablation study, it remains challenging to assess the sensitivity of the prediction.

**Suitability:**

3

---

### Official Review · Reviewer_xaRW · 2024-05-24

**Rating:** 2
**Confidence:** 3

**Summary:**

The manuscript introduces one database of images and text for controversy detection using the Chinese language. In addition, it presents one architecture for this objective.

**Strengths:**

The paper is very well written and structured leading to a manuscript of high quality. The context is in general concrete and the authors managed to comprehensively describe their technical contributions. Even if there are issues concerning the results, the latter can be considered sufficient in terms of adequacy. Reference list is more than complete and to be precise, it should be shortened as it contains some obsolete references which has no use in the manuscript.

**Limitations:**

Despite its high quality as it concerns the manuscript itself, there are some major issues that should be considered in general:
-	Whether something is controversy or not, from my perspective it is a subjective matter. Despite the correct approach of the authors to involve in the annotation process various and diverse individuals, parameter of culture plays a pivotal role in how each individual considers a thread as a controversy post. Hence, if the individuals are of the same nationality, most likely they will produce a biased dataset. Unless if we consider that the “way of thinking” and the culture of a nationality cannot affect the categorization of a post as a controversy.
-	As the proposed database and model, focuses on the Chinse language, its application is extremely limited. MMCD includes textual embeddings for the Chinese language. MVCD can detect a controversy only for the Chinese language. This limits extremely the application.
-	In line 222, the authors claimed that their approach is unique. At first, it would be expected that there will be no comparison with similar multi-modal controversy detection models which is not the case. In addition, after a short research, such an approach would be reference Mariconti, E., Suarez-Tangil, G., Blackburn, J., De Cristofaro, E., Kourtellis, N., Leontiadis, I., ... & Stringhini, G. (2019). " You Know What to Do" Proactive Detection of YouTube Videos Targeted by Coordinated Hate Attacks. Proceedings of the ACM on Human-Computer Interaction, 3(CSCW), 1-21. I suppose others could also be found.
-	There is no details provided concerning the creation of the datasets and crawling these data and more specifically for any ethics related concerns, keywords used etc.
-	No information provided how these “sentiment analysis” and its results were incorporated (Lines 442-452).
-	Many parameters were provided in Section which also refer to Figure 5. The latter should be updated with the parameters (inputs/outputs per component).
-	Based on the descriptions of each model (section 4), it is assumed that eventually the proposed framework is just a pipeline. Though, the authors did not provide any justification for selecting each approach for each pipeline component.
-	It is mentioned that the used models for text embeddings are pre-trained. Though it is not clarified on which language assuming Chinese. Correct?
-	From my perspective, experimental results and more specific their commentary should be expanded. Apart from that, the comparison is uneven as all multi-modal approaches are not actually “multi-modal” as claimed by the authors (check my previous comment). In addition, fake detection (see line 743) is not so relevant with the controversy detection so how relevant is the comparison?
-	There is confusion of the main objective of the paper. In the title, the authors focusing on the dataset while “Experimental results” focus on the proposed framework undermining the dataset and the title per se.

**Suitability:**

3

---

### Meta-Review · Area_Chair_se7f · 2024-06-28

**Recommendation:** Accept (Poster)
**Confidence:** 4

**Metareview:**

The reviewers are unanimous in their opinion that the paper should be accepted. Most of them are particularly positive about the introduced MMCD dataset, as well as the proposed MVCD approach. In addition, the reviewers agree that the paper is definitely relevant to the scope of ACM MM, clear and well structured. On the other hand, the reviewers have also expressed concerns about e.g. definition of controversy, code availability and the potential ethical issues.

Taking into account the initial reviews and the rebuttal, I believe that the paper should be accepted.